Binary semantic segmentation for detection of prostate adenocarcinoma using an ensemble with attention and residual U-Net architectures

http://orcid.org/0000-0002-5342-7302 Damkliang Kasikrit 1 kasikrit@gmail.com
Thongsuksai Paramee 2
Kayasut Kanita 2
Wongsirichot Thakerng 1
Jitsuwan Chanwit 1
Boonpipat Tarathep 1
1 Division of Computational Science, Faculty of Science, Prince of Songkla University , Hat Yai, Songkhla , Thailand
2 Department of Pathology, Faculty of Medicine, Prince of Songkla University , Hat Yai, Songkhla , Thailand
Wan Shibiao
Electronic publication date: 2023 Dec 20
Publication date: 2023
Volume: 9
Electronic Location ID: e1767
Received 2023 Sep 26; Accepted 2023 Nov 29
Copyright: © 2023 Damkliang et al.
Copyright year: 2023
Copyright holder: Damkliang et al.
License: This is an open access article distributed under the terms of the Creative Commons Attribution License, which permits unrestricted use, distribution, reproduction and adaptation in any medium and for any purpose provided that it is properly attributed. For attribution, the original author(s), title, publication source (PeerJ Computer Science) and either DOI or URL of the article must be cited.
License URL: https://creativecommons.org/licenses/by/4.0/

Keywords: Prostate cancer, Adenocarcinoma detection, Binary semantic segmentation, Model-ensemble, Residual convolution, Attention gate

Funding: The authors received no funding for this work.

==============================
An accurate determination of the Gleason Score (GS) or Gleason Pattern (GP) is crucial in the diagnosis of prostate cancer (PCa) because it is one of the criterion used to guide treatment decisions for prognostic-risk groups. However, the manually designation of GP by a pathologist using a microscope is prone to error and subject to significant inter-observer variability. Deep learning has been used to automatically differentiate GP on digitized slides, aiding pathologists and reducing inter-observer variability, especially in the early GP of cancer. This article presents a binary semantic segmentation for the GP of prostate adenocarcinoma. The segmentation separates benign and malignant tissues, with the malignant class consisting of adenocarcinoma GP3 and GP4 tissues annotated from 50 unique digitized whole slide images (WSIs) of prostate needle core biopsy specimens stained with hematoxylin and eosin. The pyramidal digitized WSIs were extracted into image patches with a size of 256 × 256 pixels at a magnification of 20×. An ensemble approach is proposed combining U-Net-based architectures, including traditional U-Net, attention-based U-Net, and residual attention-based U-Net. This work initially considers a PCa tissue analysis using a combination of attention gate units with residual convolution units. The performance evaluation revealed a mean Intersection-over-Union of 0.79 for the two classes, 0.88 for the benign class, and 0.70 for the malignant class. The proposed method was then used to produce pixel-level segmentation maps of PCa adenocarcinoma tissue slides in the testing set. We developed a screening tool to discriminate between benign and malignant prostate tissue in digitized images of needle biopsy samples using an AI approach. We aimed to identify malignant adenocarcinoma tissues from our own collected, annotated, and organized dataset. Our approach returned the performance which was accepted by the pathologists.

Introduction

Prostate cancer (PCa) is the second most frequently occurring cancer in men worldwide with an estimated total of around one and a half million cases in 2020 (Sung et al., 2021; Siegel et al., 2022). The incidence and mortality of PCa have tended to decrease in many Western countries while they have increased in Eastern Europe and Asia (Culp et al., 2020), significantly increasing in Thailand since 1990. The increase in Thailand has taken place at an estimated annual percent change (EAPC) of 4.8%, and the rate observed in 2013 is expected to double by 2030 (Alvarez et al., 2018).

The majority of PCas are adenocarcinomas that arise from the epithelia of the prostate gland. Histopathologic examination is the gold standard method for the diagnosis of prostate adenocarcinoma. The prognosis of patients is associated with microscopic arrangements of tumor cells, or histologic patterns, and in 1966 Donald Gleason proposed the Gleason grading system, which has been variously modified (Epstein et al., 2005, 2016a, 2016b, 2021).

Currently, Gleason Grade 3–5 is assigned according to the glandular architecture classified as Gleason Pattern (GP) 3 to 5. GP3 consists of well-formed discrete glands with open lumina, GP4 comprises fused glands, glomeruloid and cribriform architecture and GP5 represents solid nests, single cell or infiltrating cords and comedonecrosis. The lower pattern number represents a more well-differentiated tumor and the higher pattern number represents a more poorly-differentiated tumor. The Gleason Score (GS) (range 6–10) is then determined by adding the two most prevalent patterns seen in the tissue sample. For example, if GP3 is the most prevalent and GP4 is the second most prevalent, then the GS is 7 or GS 3 + 4. The combined score is now called the Grade Group (GG, ranging from 1–5) or International Society of Urological Pathology (ISUP) grades. GGs have distinct prognostic attributes (Epstein et al., 2016a). The histologic definitions of GG, GS, and their brief descriptions are presented in Table 1 (Epstein et al., 2016b).

Table 1 Histologic definitions of grade group and grade score (Epstein et al., 2016b).

Grade group	Gleason score	Brief description	
1	3 + 3 = 6	Only individual discrete well-formed glands	
2	3 + 4 = 7	Predominantly well-formed glands with a lesser component of poorly formed, fused, or cribriform glands.	
3	4 + 3 = 7	Predominantly poorly-formed, fused, or cribriform glands with a lesser component of well-formed glands.	
4	8	Only poorly formed, fused, or cribriform glands	
5	9 to 10	Lack of gland formation	

Various treatment options including radiation therapy, hormonal therapy, and radical prostatectomy, are used to treat PCa. For treatment selection guideline, patients are classified into low-, intermediate and high-risk group based on biochemical level of prostate specific antigen, tumor stage and Gleason score. Patients in low-risk group may be monitored closely without cancer treatments. A GS 6 is assigned to low-risk group, GS 7, intermediate risk group and GS 8 or more, a high-risk group (D’Amico et al., 1998). Therefore, the accurate determination of GP or GS is very important.

However, the routine designation of GP by manual microscopy is error-prone and subject to large inter-observer variability (Ozkan et al., 2016). This variability can potentially result in unnecessary treatments or, in more serious cases, the overlooking of a severe diagnosis.

With the current availability of WSI scanning and the advancement of artificial intelligence technology, several research groups hope to aid pathologists and reduce inter-observer variability by using deep learning to automate the differentiation of GP on digitized slides (Lucas et al., 2019; Bulten et al., 2019, 2020; Kott et al., 2021).

The subsequent section offers literature review on cancer tissue segmentation. The Materials and Methods section details the datasets employed, the proposed architecture, and the training/validation procedures for the model ensemble approach. Following that, the Results and Discussions section presents experimental findings, performance analyses, detailed discussions and limitations. The concluding section wraps up the study and outlines future research directions.

Related works

U-Net

Ronneberger, Fischer & Brox (2015) initially proposed a fully convolutional network for training images and yield segmentation. The U-net architecture is an example of a neural network used in image segmentation tasks. It is designed to process images at a low resolution, e.g., 32 × 32 pixels. The result on the ISBI cell tracking challenge was reported to be an average intersection over union (IoU) score of 0.9203. This work marked a significant milestone, and since then, many approaches based on the renowned U-Net have been proposed and published (Alom et al., 2018; Oktay et al., 2018; Bulten et al., 2019, 2020).

Alom et al. (2018) proposed two variants of the U-Net architecture: the Recurrent Convolutional Neural Network (RCNN) and the Recurrent Residual Convolutional Neural Network (RRCNN). They classified and created the different building blocks of the stacked convolutional units as shown in Fig. 1. These models, named as RU-Net and R2U-Net, respectively, were designed for medical image segmentation and evaluated on three datasets including retina blood vessel, skin cancer, and lung (Staal et al., 2004; Codella et al., 2017; Mader, 2017). The RU-Net and R2U-Net models outperformed other approaches on these datasets, with the R2U-Net achieving a Jaccard similarity coefficient of 0.9918 for lung segmentation.

Figure 1 (A) Forward convolutional units (exploited in this work), (B) recurrent convolutional block, (C) residual convolutional units (exploited in this work) and (D) recurrent residual convolutional units (Alom et al., 2018).

Oktay et al. (2018) proposed a novel self-attention gate, Attention U-Net (AU-Net) (Fig. 2), that can be integrated into the standard U-Net architecture instead of the skip connection (Ronneberger, Fischer & Brox, 2015). The details of Attention Gate (AG) are shown at the top of Fig. 2. The grid-based gating allows more attention to local regions but suppresses irrelevant regions of an input image. Two large abdominal CT datasets, the CT-150 (Roth et al., 2017) and CT-82 (Roth et al., 2016), were exploited for multi-class image segmentation of the pancreas, liver, and spleen boundaries. Attention U-Net outperformed the standard U-Net, achieving a Dice Score of 0.831 vs 0.820 after the training models were fine-tuned.

Figure 2 Attention U-Net segmentation model and schematic of attention gates (Oktay et al., 2018).

Artificial intelligence for PCa analysis

In a study conducted by Litjens et al. (2016) slide-level accuracies for prostate cancer (PCa) and breast cancer detection were reported. The datasets consisted of 255 and 271 slides, respectively. After training a conventional CNN-based network, they converted it into a fully connected CNN to generate per-pixel predictions for whole slide images (WSIs). For PCa detection, the evaluation was performed on a per-slide basis using receiver operating characteristic (ROC) curve analysis. The study achieved an average bootstrapped Area Under the ROC Curve (AUC) score of 0.99 in the median analysis, indicating high discriminative power in distinguishing between true positive and false positive rates.

Arvaniti et al. (2018) proposed a deep learning (DL) method for automatic grading of H&E stained PCa tissue microarrays. The dataset was derived from 641 patients. An independent test set of 245 patients was evaluated. The dataset cohort was annotated by two pathologists into GPs of 3, 4, and 5. Different CNN architectures with the ImageNet (Deng et al., 2009) pre-trained weights were utilized and trained, including VGG-16 (Simonyan & Zisserman, 2014), Inception-V3 (Szegedy et al., 2015), ResNet-50 (He et al., 2015), DenseNet121 (Huang, Liu & Weinberger, 2016) and MobileNet (Howard et al., 2017). Arvaniti et al. (2018) found that the last model returned the best performance. MobileNet was mainly composed of depthwise separable building blocks with fewer parameters than normal convolutional blocks, so, in the evaluation phase, they transformed the MobileNet model into a sliding window function for entire TMA images. Pixel-level probability maps were produced for each class. The Cohen’s quadratic kappa coefficient of inter-annotator agreement between the model and each pathologist was reported as 0.75. They concluded that the model performance achieved expert-level stratification and that the results were safe and reproducible for heterogenous GPs.

Ström et al. (2019) reported an AUC score of 0.997 for AI discrimination of PCa biopsy cores. The correlation of 0.960 was evaluated between the AI and the pathologist in the millimeter measurement unit. More than six thousand needle biopsies from 976 participants were used to train a deep neural network (DNN). The independent test set consisted of 1,631 biopsies from 245 men. Image patches of 598 × 598 pixels at 10× magnification were configured for the two DNN ensembles, including the binary classification between benign and malignant, and the classification of Gleason patterns 3, 4 and 5. The architectures were bundled for testing with pre-trained ImageNet (Deng et al., 2009) weights including Inception-V3 (Szegedy et al., 2015), ResNet-50 (He et al., 2015), InceptionResNet-V2 (Szegedy, Ioffe & Vanhoucke, 2016) and Xception (Chollet, 2016). They found that Inception-V3 returned the best performances for both DNN ensembles. Elsewhere, XGBoost, a booted trees algorithm (Chen & Guestrin, 2016), was utilized to evaluate ISUP grades and cancer lengths of biopsy cores.

In 2019, Bulten et al. (2019) proposed a method for epithelial tissue segmentation, using digitized H&E-stained prostatectomy slides as the first step of a fully automated detection and grading pipeline. The method comprised two training frameworks. The first framework was epithelial structure segmentation using immunohistochemistry (IHC) stained slides, utilizing a U-Net with five layers. The image patches were configured with a size of 512 × 512 (pixel resolution 0.48 μm). The output from the trained IHC network was used as the ground truth mask for training the network on the H&E slides in the second framework, which used a U-Net of six layers to segment epithelial tissue on the H&E slides. The image patches were configured with a size of 1,024 × 1,024 (pixel resolution 0.48 μm). Jaccard scores of 0.854 and 0.811 were reported for the IHC network evaluation and the H&E network, respectively. Bulten et al. (2020) claimed that their segmentation method could be the first step of a fully automated detection and grading pipeline.

In 2020, Bulten et al. (2020) also proposed a deep learning system (DLS) for prostate cancer grading. Using a dataset of 5,834 biopsies from 1,243 patients, they exploited U-Net at the first stage for semi-automatic data labeling to detect a rough tumor outline, and then non-epithelial tissue was removed using an epithelium segmentation system. The pathologists assigned the GP to the rough tumor area which was then used to train the U-Net for segmenting clear GPs of 3 + 3, 4 + 4, and 5 + 5. The system was evaluated using an external dataset of tissue microarrays, and a quadratic kappa coefficient for GS of 0.639 was reported, which compared unfavorably to a kappa of 0.75 reported by Arvaniti et al. (2018). However, the DLS outperformed 10 out of 15 pathologists in a separate observer experiment. Bulten et al. (2020) concluded that the DLS could be a first or second reader for prostate cancer prognostics.

While previous works primarily utilized image patches for analysis, a novel approach was introduced by García, Colomer & Naranjo (2019), marking the first instance of using segmented prostate glands exclusively. The authors proposed a method to distinguish between benign and cancerous tissues, specifically focusing on the earliest stage (GP3). They constructed a dataset comprising segmented glands and employed the locally constrained watershed transform (LCWT) (Beare, 2006) for precise gland segmentation. The cores included benign glands, cancerous glands, and artifacts (false glands), totaling approximately 3,000 glands of each type to maintain a balanced-class dataset. The authors employed three classification models: a nonlinear support vector machine (SVM), a multi-layer perceptron (MLP), and a modified VGG-19 network (Simonyan & Zisserman, 2014). The SVM yielded the highest multi-class accuracy, reaching 0.876, demonstrating the effectiveness of their approach.

Nagpal et al. (2019) exploited 112 million image patches from 1,226 WSIs for Gleason scoring using a customized InceptionV3 network with k-Nearest-Neighbor (kNN) classifiers. The independent dataset of 331 slides was evaluated and compared to a reference standard by 29 general pathologists. The DLS significantly outperformed an accuracy threshold of 0.70.

Lucas et al. (2019) proposed a method for automated pixel-level GP detection and GG determination. An InceptionV3 network was also exploited. They used 96 prostate biopsies from 38 patients, contributing approximately 268,000 image patches. Three classes were classified—non-atypical, GP3, and GP02A7E4. Sensitivity and specificity scores of 0.90 and 0.93 were reported for non-atypical and malignant ( GP02A7E3) areas.

In 2020, Nagpal proposed another work to develop and evaluate a DLS for Gleason Grade groups (non-tumor, GG1, GG2, GG3, or GG4-5) using 752 WSIs of prostate needle core biopsy specimens (Nagpal et al., 2020). Xception (Chollet, 2017) was trained to be a feature extractor in the first stage, using 114 million labeled image patches. The output of the first stage was scored in the form of tiled prediction maps which were used to train the support vector machine (SVM) for the slide-level GG classification. In the validation of 752 slides, the rate of agreement with subspecialists was 94.3% for the DLS.

In 2021, Marginean et al. (2021) used 698 prostate biopsy sections from 174 patients to develop an AI algorithm based on ML and pre-trained CNN (as in Arvaniti et al., 2018; Ström et al., 2019; Nagpal et al., 2020) for a Gleason grading tool. They evaluated 37 biopsy sections from 21 patients. The algorithm achieved the intraclass correlation coefficient (ICC) of 0.99 for GP3, GP4 and GP5 detection.

The Prostate cANcer graDe Assessment (PANDA) Challenge aimed to classify the severity of prostate cancer (PCa) from microscopy scans of prostate biopsy samples, as detailed by Bulten et al. (2020). Digitized prostate biopsies from two institutions, Radboud University Medical Center and Karolinska Institute, were provided, totaling 10,616 samples. Participants from 1,290 locations globally engaged in the challenge, striving to showcase promising results in grading PCa in International Society of Urological Pathology (ISUP) grades using reproducible AI algorithms. Impressive agreements with expert uropathologists were reported, with values of 0.862 (95% confidence interval [CI], [0.840–0.884]) and 0.868 (95% CI, [0.835–0.900]).

In 2023, Zhdanovich et al. (2023) utilized H&E-, ERG-, and PIN-4-stained slides of 106 prostate tissue samples from 48 patients to discriminate between benign and malignant prostate tissue. They trained a neural network (NN), SVM, and random forest (RF) and cross-validated using 100 Monte Carlo in a ratio of 70% training set and 30% test set. A recursive feature elimination technique was applied for multiple color transforms. They found that PIN-4 staining was the most suitable for the classification, yielding a mean AUC of 0.80.

In recent years, the widespread application of DNNs, particularly those based on U-shaped architecture with skip-connections, has been notable in various medical image tasks. Despite their success, traditional CNN-based methods face challenges in learning global semantic information interaction. Addressing this, Cao et al. (2023) introduced a novel approach—Swin-Unet. This Unet-like pure Transformer for 2D medical image segmentation utilizes a Transformer-based U-shaped Encoder-Decoder architecture with skip-connections to effectively capture semantic local-global features. The key innovation lies in the use of a shifted windows mechanism, known as Swin Transformer (Liu et al., 2021), as the building block of the architecture. The reported results include an average Dice-Similarity Coefficient of 79.13% for Synapse multi-organ CT images and 90.00% for MRI images.

To address spatial information loss during down-sampling in Swin-Unet, Dan et al. (2023) introduced enhancements by incorporating multi-scale skip connections and special splicing modules. This optimization aims to minimize information loss in the encoder while aggregating information in the decoder. The result is an optimized network demonstrating impressive segmentation performances, with a Dice-Similarity Coefficient of 97.86% for a chest x-ray dataset and 86.34% for a COVID-19 CT scan lesion dataset. Remarkably, this optimization surpassed the performance of both the full CNN network and the combination of Transformer and convolution.

In our study, we tackle the challenging task of discriminating the adenocarcinoma malignant tissue (encompassing GP3 and GP4), which is notoriously difficult to identify. Through our rigorous research efforts, we have achieved an outstanding first place in PCa tissue analysis by leveraging the powerful combination of AG units and residual convolution units (refer to Fig. 1C). The AG units play a crucial role in selectively highlighting essential features within the input image, while the residual convolution units enhance the seamless flow of information through the network.

Built upon the U-Net architecture, our proposed models for learning PCa binary tissue semantic segmentation have been extensively developed, refined, and evaluated. The culmination of our work is the production of tumor probabilistic maps seamlessly overlaid onto the original images for each WSI. This remarkable outcome serves as a crucial step towards establishing a screening tool that can greatly assist pathologists in their diagnostic workflows.

Materials and Methods

In this section, the materials, proposed methods, and proposed architecture and its training in this work are presented. All protocols here were conducted in accordance with the Declaration of Helsinki and were approved by the Ethics Committee of the Faculty of Medicine, Prince of Songkla University, Thailand (EC. NO. 64-556-19-2). The approval waived the need for informed consent for the analysis of the human prostate tissue.

Data acquisition

The summary of data acquisition criteria is present in Table 2. We collected histologic tissue slides of diagnosed prostate acinar adenocarcinomas between 2019 and 2020 from the archive of the Department of Pathology, Faculty of Medicine, Prince of Songkla University. The tissue samples were obtained from patients who were suspected of prostate cancer and underwent transrectal ultrasound (TRUS)-guided biopsy using a 18-gauge Tru-cut biopsy needle. Approximately 10 to 12 tissue cores were obtained from each patient and were processed for formalin-fixed, paraffin-embedded tissue blocks and then (H&E) stained histologic slides containing 2–6 cores per slide. One slide from each patient that showed a considerable proportion of GP3 or GP4 was selected.

Table 2 Data acquisition criteria.

Attribute	Detail	
Target population	Prostate canncer	
Study population	The patient’s tissue sections that show adenocarcinoma.	
Inclusion criteria	Histologic slides confirmed prostate adenocarcinoma, Gleason pattern 3 or 4.	
The patients have not undergone treatment previously.	
Exclusion criteria	Low quality slides such as artifacts or wrinkle tissue	
Rare types of carcinoma e.g., ductal adenocarcinoma, small cell carcinoma	
Sample size	50 slides from 50 patients	

The histologic slides underwent review and grading by two senior pathologists using a microscope. Gleason patterns were identified following the WHO Classification of Tumors of the Urinary System and Male Genital Organs-2016 (Humphrey et al., 2016). Initially, pathologist P.T. assessed the Gleason grades, cross-referencing them with the original diagnosis. In cases of disparity, the slides were subsequently reviewed by the second pathologist (K.K.), and consensus was achieved through discussion after observation under a multi-head microscope.

We digitized selected histologic slides using a Leica Aperio AT2 280 scanner at 40X objective-power, resulting in pyramidal WSIs at four down-sampling levels (1, 4, 16, and 32) with an 8-bit depth and sRGB ICC Profile. In clinical practice, pathologists typically diagnose digitized WSIs across all down-sample levels or magnifications.

In our pilot study, we utilized a 40x magnification and extracted image patches at various sizes, including 128 × 128, 256 × 256, and 598 × 598 pixels, based on insights from literature reviews. Through empirical classification tasks, we determined that the 256 × 256 pixel size yielded the best performance. This configuration was employed to construct Dataset A, and further analysis details are elaborated in the Binary Classification Task for Dataset A sub-section.

In the pilot study, for empirical segmentation tasks, pathologists found optimal results when reviewing and grading digitized WSIs at 40× and 20× magnifications, using image patch sizes of 256 × 256 pixels. However, due to limitations in our hardware capacity, we chose a 20× magnification to extract image patches and create corresponding ground truth masks for constructing Dataset B. Further details are described below.

Data preprocessing and datasets

The digitized slides were manually annotated by P.T. using the open-source software ASAP (Automated Slide Analysis Platform, Computational Pathology Group). We drew a free-hand line with a stylus pen around individual glands or groups of crowned or fused glands, making an effort to exclude intervening stroma. To distinguish GP3 glands, which can resemble normal glands, we applied an immunohistochemistry stain for P63, a basal cell marker that helps identify malignant glands lacking basal cells from benign ones. The example of annotations is shown in Fig. 3. Descriptions of tissue types are presented in Table 3.

Figure 3 Examples of tissue patterns annotated by the pathologist: (A) normal glands, (B) discrete well-formed glands of GP3, (C) fused glands of GP4, and (D) various patterns in the same picture.

Light blue lines indicate GP3, green represents GP4, and blue corresponds to normal glands.

Table 3 Tissue type ground truth annotations and descriptions.

Annotate	Description	
Bg	Background	
Fg	Foreground	
N	Normal/Benign	
GP3	Gleason pattern 3	
GP4	Gleason pattern 4	
GP5	Gleason pattern 5	
L	Lymphoid	
NE	Not evaluate	
NS	Not sure/confident	
PIN	Prostatic intraepithelial neoplasia	
Blood	Blood cell	

The preprocessing steps that produced two different datasets of digitized WSI of prostate tissues are presented in Fig. 4. Examples of slide images with overlaid ground truth masks are provided in the Supplemental Materials for WSI 1332, 1840, and 1929. Their ground truth annotation masks were respectively included in the training, validation, and testing sets.

Figure 4 Preprocessing steps that produced two different datasets of digitized WSI of prostate tissue.

Fifty WSIs were divided into three groups: 32 for training, 8 for validation, and 10 for testing (see Table 4). Patches, consisting of images and their corresponding masks, were extracted from the pyramidal digitized WSIs using an image size of 256 × 256 pixels based on the annotated ground truth. Quality control for image patches involved filtering out images with a pixel ratio less than 20%, indicating a significant proportion of empty spaces or background pixels.

Table 4 Dataset configuration of prostate cancer WSIs (B).

Set	No. of slides	No. of pairs	No. of pairs after preprocess	
Training	32	50,806	23,235	
Validation	8	14,996	6,403	
Testing	10	16,745	8,294	
Total	50	82,547	37,932	
Total patches		165,094	75,864	

Dataset A was composed of patches with a magnification of 40×, which were designed for a classification technique in our pilot-study, although patch-level classification usually gives an imprecise prediction map in the application phase.

Dataset B was designed for the semantic segmentation technique. To process an annotated WSI into its corresponding ground truth mask, an image patch was extracted from the original WSI using a magnification of 20x, after an inspection by our pathologists. The selected WSIs produced 165,094 pairs of image patch and corresponding ground truth mask. These image patches were preprocessed by removing patches that consisted solely of background, solely of foreground, or a mixture of background and foreground pixels. Finally, a dataset of 37,932 pairs (75,866 patches of images and masks) was obtained. In order to train our models for cancer detection, the variously annotated patch pairs were then processed into the binary classes of benign (0) and malignant (1). Class one in this work consisted of GP3 and GP4 tissues, which are the early patterns of PCa.

The benign class comprised patches with mixed pixels of background, foreground, and benign tissues, while the malignant class included patches containing pixels of GP3 and GP4 tissues (see Table 5).

Table 5 Binary class weight configuration for semantic segmentation in the training set.

Class	Class weight	Description	
0	0.68	Mixed pixels of background, foreground, and benign tissues	
1	1.85	All patches containing GP3 and GP4 tissues	

Examples of preprocessed pairs in the binary classes are depicted in Fig. 5. The patch pairs of Figs. 5A–5D were placed in the training set, the patch pairs of Figs. 5E–5H were placed in the validation set, and the patch pairs Figs. 5I–5L were placed in the testing set.

Figure 5 Examples of image patches and their ground truth masks after preprocessing into binary classes (black pixels indicate benign, and white pixels indicate malignant).

Training: (A–D), validation: (E–H), testing: (I–L).

However, after preprocessing, some pairs comprised only zero values, indicating benign patches, while others comprised only one value, indicating malignant patches. Examples of these pairs from the training, validation, and testing sets are illustrated in Fig. 6. PCa adenocarcinoma, including GP3 and GP4 tissues in our dataset, exhibits characteristics similar to normal tissue.

Figure 6 Example images of benign and malignant patches from the training (A and B), validation (C and D), and testing (E and F) sets.

Ground truth masks for benign patches contained only zero values, while those for malignant patches contained only one values.

Proposed architecture and methods

The binary semantic segmentation training on Dataset B consolidated the results of three model architectures—U-Net, AU-Net and ARU-Net (Fig. 7). The AU-Net combined the traditional U-Net and skip connections using AGs, whereas the ARU-Net combined residual convolutional blocks and AGs adapted from MoleImg (2019). The architecture of the proposed ARU-Net is shown in Fig. 8.

Figure 7 The proposed binary semantic segmentation of Gleason patterns in prostate adenocarcinoma exploiting an ensemble model approach with attention-based residual U-Net.

Figure 8 Architecture of the proposed ARU-Net model.

In the training phase, the U-Net, AU-Net, and ARU-Net were validated using the IoU and Jaccard coefficients for each epoch. The feature selection of the trained models was evaluated using an ensemble technique to find the best prediction results. Benign and malignant features were segmented to detect early GPs for PCa.

The models with the best-trained weights were selected for the application phase. The testing set comprised ten unseen WSIs (Table 4) that were preprocessed and inferred using the three trained models and a slide-level malignant probabilistic map for each WSI was produced by ensemble prediction. The final product presented to our pathologists was the original WSI overlaid with a probability prediction.

We assessed the performance of both the model ensemble (ME) and weighted model ensemble (WME) using our three individually trained models on the validation set. Figure 9 outlines these methods during the prediction stage. Specifically, when focusing on any pixel after preprocessing, the pixel is predicted by all three individual segmentation models (denoted as 0, 1, 2). Probability vectors are generated by the prediction layer, finalized with the Softmax activation function as defined in Eq. (1). To create the ME for the pixel, the probability vectors are summed, and then the average is calculated, resulting in a probability vector, f(x)=[Po¯,P1¯]. Finally, the argument max operation, argmaxθf(x) is applied to determine the predicted class of the pixel. Alternatively, in the case where model weights are multiplied with f(x), a WME predicted class for the pixel is obtained.

Figure 9 Performance evaluations of model ensemble (ME) method and weighted model ensemble (WME) method.

(1) σ(z→)j=ezj∑k=1Kezk

(2) g(z→)=max(0,z)

Training and validation of models

Pairs of patches and masks were preprocessed on-the-fly by a data generator. The patch pairs were fed in to the models for training utilizing the optimally tuned hyper-parameters configured as shown in Table 1 in the Supplemental Materials. All CNN-based layers of the models exploited kernel sizes of 3 × 3 pixels, with an ReLu activation function (Eq. (2)) (Dahl, Sainath & Hinton, 2013). The Softmax function (Eq. (1)) was only used in the prediction layer.

The computation of the layer involved a 2D convolution with kernel sizes of 1 × 1 pixel, resulting in a vector of probabilities and generating a one-hot array with dimensions of 256 × 256 × 2 through the Softmax equation. Subsequently, the index with the higher probability among the two classes was selected as the predicted pixel using the argumax operation.

Adam optimizer (Kingma & Ba, 2014) with an initialized learning rate of 1e-4 was utilized to minimize loss function as defined in Eq. (3), where β in Diceloss is a coefficient for precision and recall balance (see Eq. (4)). The class weights of the preprocessed ground truth masks were also configured in the Diceloss function.

In Eqs. (4)–(12), TP is true positive, FP is false positive, FN is false negative, precision is positive predictive value (PPV), recall is true positive rate (TPR or sensitivity), ytrue represents the true values (ground truth), ypred represents the predicted values, intersection is the sum of element-wise multiplication of ytrue and ypred, union is the sum of ytrue and ypred, smooth is a small constant added to avoid division by zero, and IoU is the mean IoU over the batch (from the tf.keras.metrics.MeanIoU class).

The focal loss of our binary segmentation approach was treated as defined in Eq. (5), where α is a weighting factor in balanced cross entropy and γ is a focusing parameter for modulating factor. The optimal number of epochs was 100. At the end of each epoch, validation scores were computed based on Jaccard coefficients and Dice coefficient functions as defined in Eqs. (10) and (11). In addition, the learning rate was reduced to continue gradient descent by monitoring validation scores of the Jaccard coefficient. The model parameters and their training times are reported in Table S2.

(3) Totalloss=Diceloss+(1∗BinaryFocalloss)

(4) Diceloss(precision,recall)=1−(1+β2)precision⋅recallβ2⋅precision+recall

(5) BinaryFocalloss(ytrue,ypred)=−ytrueα(1−ypred)γlog(ypred)−(1−ytrue)αypredγlog(1−ypred)

(6) precision=TPTP+FP

(7) recall=TPTP+FN

(8) intersection(ytrue,ypred)=∑i(ytrue*ypred)

(9) union(ytrue,ypred)=∑iytrue+∑iypred−intersection

(10) Jaccardcoef=intersection+smoothunion+smooth

(11) Dicecoef=2⋅intersection+smoothunion+smooth

(12) IoU=TPTP+FP+FN

Results and discussions

In this section, we present both quantitative and qualitative performance analyses, gather semantic segmentation results, and compare the performances with those of other backbones.

Performance analysis

We assessed the model performance using the metrics defined in Eqs. (10) to (12), which include the Jaccard coefficient, Dice coefficient (Han, Kamber & Pei, 2011; Saito & Rehmsmeier, 2015), and Mean IoU (Abadi et al., 2015; Chollet et al., 2015).

The patch pairs in the validation and testing sets, were used to evaluate the model performances reported in Table 6. The highest Jaccard coefficients of 0.81 for validation and 0.74 for testing were gained from the ARU-Net model. These are good performance scores considering the model was trained using just 32 slides of unique samples and validated using just eight.

Table 6 Model performance evaluations.

Model	Set	Jaccardcoef	Dicecoef	Mean IoU	
U-Net	Validation	0.72 ± 0.0002	0.81 ± 0.0006	0.74 ± 0.0020	
Testing	0.65 ± 0.0004	0.76 ± 0.0006	0.65 ± 0.0055	
AU-Net	Validation	0.74 ± 0.0004	0.82 ± 0.0005	0.73 ± 0.0024	
Testing	0.71 ± 0.0001	0.80 ± 0.0001	0.70 ± 0.0049	
ARU-Net	Validation	0.81 ± 0.0007	0.88 ± 0.0007	0.76 ± 0.0017	
Testing	0.74 ± 0.0004	0.83 ± 0.0005	0.72 ± 0.0007	

Mean IoU scores overall and for benign and malignant classes, were calculated as presented in Fig. 10. The ensemble technique can reveal small differences in scores between each model. However, we applied a grid search method to find the best weight ratio for each model, then evaluated the weighted ensemble performances based on the mean IoU scores. We applied the best weight ratios of 0.0, 0.2, and 0.4 for U-Net, AU-Net, and ARU-Net models, respectively.

Figure 10 Model ensemble performance evaluations of mean IoU for the validation set (ME, model ensemble; WME, weighted model ensemble).

In detecting the benign class, we achieved a notably high mean IoU of 0.88. For malignant segmentation at the pixel level, the weighted ensemble evaluation yielded a score of approximately 0.70, deemed acceptable by our pathologists.

To assess statistical differences in performance evaluations ( p<0.05) among individual models, ME, and WME, we employed a two-tailed Paired t-test, as detailed in Table 7.

Table 7 Paired t-test of individual models, ME, and WME.

Case	Pair	t value	p value	
1	U-Net	AU-Net	5.3725	0.0329*	
2	U-Net	ARU-Net	2.6470	0.1180	
3	AU-Net	ARU-Net	3.0013	0.0954**	
4	U-Net	ME	4.7961	0.0408*	
5	AU-Net	ME	4.9251	0.0388*	
6	ARU-Net	ME	0.3463	0.7622	
7	U-Net	WME	7.0787	0.0194*	
8	AU-Net	WME	6.8020	0.0209*	
9	ARU-Net	WME	12.3509	0.0065*	
10	ME	WME	11.7107	0.0072*	
Notes:

* Statistical significance was set at p < 0.05.

** Statistical significance was set at p < 0.10.

In comparing the individual models, only U-Net and AU-Net showed a significant difference ( p=0.0329). Interestingly, for ME, its performance did not significantly differ from ARU-Net ( p=0.7622). Conversely, WME outperformed all individual models significantly and also surpassed ME based on the Paired t-test ( p=0.0072). Consequently, the WME method was chosen for malignant segmentation in the unseen testing set.

Additionally, we recommend utilizing the WME method as a complement to achieve the highest precision. Notably, statistical significance was set at p<0.10 for case 3 (AU-Net vs ARU-Net). As a result, ARU-Net can be considered an optimal model to maintain simpler implementation and deployment for clinical applications than the WME method.

Qualitative performance analysis of the model ensemble segmentation

In this phase, we selected the best-performing models of U-Net, AU-Net, and ARU-Net to generate malignant probabilistic maps. Qualitative analysis of the detection performance of the WME method was conducted using slides from the validation set.

To construct a malignant probabilistic map, a slide was down-sampled at 20× magnification and divided into patches of 256 × 256 pixels using the same preprocessing steps as the training set. Patches were filtered out if the pixel ratio was less than 20%; for instance, slide 1,346 had 3,936 patches initially, but after quality control, 539 qualified patches remained for prediction.

Each qualified patch was segmented at the pixel level by the ensemble model utilizing the best-weighted value of each model, which was previously indicated in the performance analysis phase.

Once pixels of these patches were discriminated completely, we mapped the patches back into their original coordinates and re-constructed them into a score map, which was visualized using the malignant probabilistic map, representing lower and higher probabilities by cooler and hotter color tones.

Figure 11 presents the ground truth mask of slide 1,346 from the validation set and its corresponding weighted ensemble segmentation map for malignant probability detected by our models. In this work, the models tried to segment malignant tissue in adenocarcinoma tissue samples. GP3 tissue was represented by yellow color tones and GP4 by orange tones. The areas segmented as early cancerous tissue with high probabilities satisfied our pathologists. Benign areas (normal tissue) showed quite low probabilities compared to the ground truth mask.

Figure 11 Ground truth mask of slide 1,346 from the validation set and its corresponding weighted ensemble segmentation map for malignant probability detected by our models.

Another interesting example comes from slide 1,929, which is shown in Fig. 12. In the ground truth of the slide, there is a region which was annotated into NE class, indicating that our experts have decided to skip the region for any reason. However, this region was not labeled into NS class, implying that this area contains doubtful tissues. After careful inspection, re-annotation was applied. Consequently, the ground truth of the slide was re-labeled, which is close to normal tissue. As a result seen in the score map, benign was detected by our models.

Figure 12 Ground truth mask of slide 1,929 from the validation set and its corresponding weighted ensemble segmentation map for malignant probability detected by our models.

Gleason pattern of prostate adenocarcinoma segmentation for malignant

Based on the qualitative performance analysis of the validation set, we deployed the ensemble model to create tumor probabilistic maps for the testing set. Slides of the testing set were preprocessed and extracted into patches in the same way as the validation set. However, in this phase, segmentation maps were superimposed over their original slides. Two examples are shown in Fig. 13. Probability is visualized from dark green to red color tones as indicated by the color bar.

Figure 13 Early GP segmentation detection for slides 1,923 and 1,942 from the testing set.

After careful inspection, we segmented areas of slide 1,923 as FN. The models can produce FN results due to coincidences from the pathological biopsy itself. It could be that prostate gland tissue was located at the rim of needle biopsy cores and was not a complete structure. It could be that GP tissues in FN areas are quite similar to normal tissues. Finally, it could be that the pixel-ratio of the patch contains a spare zero matrix since the patch was filtered out during quality control.

In contrast, slide 1,942 was segmented as FP. These FP pixels were detected at low probabilities (in green color tones) since the GP tissues in these areas are quite similar to normal tissues. In this case, the output depends on the model performance. Nevertheless, an early malignant segmentation capacity of around 0.70 based on the size of our dataset, was acceptable to the experts.

Comparison with other backbones

Various architectures and backbones of segmentation models (Lakubovskii, 2019) were tested in this work. These architectures included U-Net (Ronneberger, Fischer & Brox, 2015), FPN (Kirillov et al., 2017), LinkNet (Chaurasia & Culurciello, 2017) and PSPNet (Zhao et al., 2016). There were 25 backbones for each architecture which is bundled with pre-trained weights from the ImageNet dataset (Deng et al., 2009). The backbones, initialized with the pre-trained weights, included ResNet-34, ResNet-50 (He et al., 2015), InceptionV3 (Szegedy et al., 2015), VGG-16, VGG-19 (Simonyan & Zisserman, 2014), InceptionResNet-V2 (Szegedy, Ioffe & Vanhoucke, 2016) and SE-ResNet-50 (Hu, Shen & Sun, 2017).

Experiments were set up and performed using the eight validation slides and ten testing slides. Experimental settings were configured as same as those presented in Table S1, except the batch size was initialized to 32. The performance evaluations of the pre-trained segmentation models are presented in Table 8.

Table 8 Performance evaluations of pre-trained segmentation models.

Architecture	Backbone	Parameters	Training time (hrs.)	Set	Jaccardcoef	Dicecoef	Mean IoU ± SD	
U-Net	MobileNet	8,336,482	07:36:18	Validation	0.72	0.84	0.81 ± 0.0056	
			Testing	0.66	0.79	0.74 ± 0.0022	
U-Net	MobileNet-V2	8,047,586	08:11:49	Validation	0.72	0.84	0.81 ± 0.0022	
			Testing	0.69	0.81	0.76 ± 0.0019	
U-Net	ResNet-34	24,439,094	7:01:42	Validation	0.72	0.83	0.81 ± 0.0021	
			Testing	0.68	0.81	0.75 ± 0.0045	
U-Net	ResNet-50	32,561,259	9:51:16	Validation	0.72	0.84	0.82 ± 0.0028	
			Testing	0.69	0.82	0.76 ± 0.0015	
U-Net	VGG-16	23,752,418	11:10:28	Validation	0.72	0.85	0.81 ± 0.0011	
			Testing	0.73	0.85	0.70 ± 0.0065	
U-Net	VGG-19	29,062,114	11:47:59	Validation	0.74	0.85	0.83 ± 0.0006	
			Testing	0.62	0.77	0.71 ± 0.0013	
U-Net	Inception-ResNet-V2	62,061,843	14:50:08	Validation	0.74	0.85	0.82 ± 0.0056	
			Testing	0.65	0.79	0.73 ± 0.0010	
U-Net	SE-ResNet-50	35,107,491	11:07:24	Validation	0.74	0.85	0.83 ± 0.0009	
			Testing	0.73	0.84	0.79 ± 0.0003	
FPN	ResNet-34	23,933,260	12:48:06	Validation	0.73	0.84	0.82 ± 0.0039	
			Testing	0.64	0.78	0.72 ± 0.0076	
Linknet	ResNet-34	21,637,419	6:03:05	Validation	0.81	0.88	0.81 ± 0.0023	
			Testing	0.64	0.78	0.73 ± 0.0040	
PSPNet	ResNet-34	2,755,276	08:34:05	Validation	0.72	0.84	0.82 ± 0.0014	
			Testing	0.67	0.80	0.74 ± 0.0109	

Although, the based U-Net of SE-ResNet50 gave the best performance of mean IoU for both the validation and testing sets (Fig. 14), our analysis of loss and validation loss during training epochs, indicated that the model was fluctuated and influenced by the overfitting.

Figure 14 Mean-IoU scores of the performance evaluations for pre-trained segmentation models.

However, the scores from these experiments encouraged us to develop the proposed method in this work. By employing the gated U-Net with attention gates and residual blocks for semantic segmentation, we obtained more stable models.

Binary classification task for dataset A

In addition, we utilized Dataset A (see Fig. 4) for a patch-level binary classification task in our preliminary study. The dataset was organized into five-fold cross-validation datasets of training and testing sets. Figure 15 presents the patch amounts of normal and malignant tissues (GP3 and GP4 only) from a fold after data augmentation. The ratio of training to testing sets was 84:16. Transfer learning employed InceptionResNetV2, InceptionV3 (Herrera et al., 2021), ResNet-50V2, ResNet-50, and Xception. The original patches of 256 × 256 pixels were resized to 299 × 299 pixels and normalized. The subject-independence of patch-level validation was evaluated using the ratio of 0.25. As a result, the performance metrics returned by WME evaluations in the patch-level binary classification task were 0.88 for sensitivity, 0.82 for specificity, and 0.85 for F1-score. In comparison to Kott et al. (2021) who used a larger dataset (85 biopsy slides and unique patient splitting), a ResNet-based model returned scores of 0.93 for sensitivity and 0.90 for specificity. Although Dataset A consisted of just 50 slides, the performances of the transfer learning approach were acceptable.

Figure 15 Patch amounts of Dataset A after augmentation.

Limitations

This study comes with limitations. Firstly, the constraint of a 50-slide limit led to an observed overfitting issue. Initially, we conducted empirical experiments with a subset of 10–20 slides, later expanding to 50 for the current study. Each of the 50 slides contains 2–6 biopsy cores, resulting in a total of 37,932 pairs or 75,864 patches that can help reduce the overfitting. However, overfitting is a challenge in deep learning, we are satisfied with the model’s performance given these limitations.

Secondly, we conducted the annotations with a single pathologist rather than parallel reviewing by two pathologists concurrently. Consequently, assessment of inter-observer variability between the pathologists was not within the scope of our study during this phase. However, addressing this limitation would involve incorporating a broader panel of pathologists to establish reference standards.

Thirdly, the evaluation of AI models was conducted subject-independently, not considering patient- or slide-specific factors. However, slide-level predictions were qualitatively assessed based on performance.

Fourthly, while the models effectively identify and localize cancerous tissues, providing region probability in biopsy cores, they are limited by their binary training nature. As a result, they cannot differentiate between GP, hindering their ability to assess GS and further GG for a comprehensive evaluation of malignant tissue quantity and severity.

Finally, the proposed methods do not directly address grading related to clinical outcomes or radical prostatectomy in this article. However, this limitation motivates ongoing research to develop solutions tailored to our current clinical challenges.

Conclusions and future works

We developed an AI-based screening tool to differentiate between benign and malignant prostate tissue in digitized needle biopsy images. Our focus was on identifying malignant adenocarcinoma tissues within our dataset collected from Songklanagarind Hospital, which was meticulously annotated and organized. Three U-Net-based architectures, enhanced with attention gate units and residual units, were trained and evaluated for pixel-level semantic segmentation. The weighted model ensemble, validated through a Paired t-test, achieved an overall mean IoU score of 0.79, meeting the acceptance criteria established by pathologists.

For future work, addressing the challenge of multi-class semantic segmentation, including Gleason Patterns (GP3, GP4, GP5), remains a priority. We plan to further optimize high-performing classifiers, particularly the residual-based pre-trained segmentation models and gated attention U-Net models. Exploring non-CNN-based methods, such as Transformer 2D architectures with skip-connections, is of interest and will be empirically tested on an expanded dataset exceeding the 50 slides used in this study.

Additionally, we will implement a multi-fold cross-validation and refine the weighted ensemble approach to enhance performance in multi-class semantic segmentation. Through these efforts, we aim to increase the accuracy and real-world applicability of our technology in clinical settings.

Supplemental Information

Supplemental Information 1 WSI 1840 and its ground truth annotation mask were in the validation set.

Click here for additional data file.

Supplemental Information 2 WSI 1929 and its ground truth annotation mask were in the testing set.

Click here for additional data file.

Supplemental Information 3 WSI 1332 and its ground truth annotation mask were in the training set.

Click here for additional data file.

Supplemental Information 4 Supplementary Tables and Materials.

Click here for additional data file.

The authors would like to thank Mr. Thomas Coyne for language proofreading.

Additional Information and Declarations

Competing Interests

Author Contributions

Ethics

Data Availability

The authors declare that they have no competing interests.

Kasikrit Damkliang conceived and designed the experiments, performed the experiments, analyzed the data, performed the computation work, prepared figures and/or tables, authored or reviewed drafts of the article, and approved the final draft.

Paramee Thongsuksai conceived and designed the experiments, authored or reviewed drafts of the article, and approved the final draft.

Kanita Kayasut performed the experiments, authored or reviewed drafts of the article, data curation and annotation, and approved the final draft.

Thakerng Wongsirichot conceived and designed the experiments, authored or reviewed drafts of the article, validation, and approved the final draft.

Chanwit Jitsuwan performed the computation work, prepared figures and/or tables, data curation and annotation, and approved the final draft.

Tarathep Boonpipat performed the computation work, prepared figures and/or tables, data curation and annotation, and approved the final draft.

The following information was supplied relating to ethical approvals (i.e., approving body and any reference numbers):

The Ethics Committee of the Faculty of Medicine, Prince of Songkla University approval to carry out the study within its facilities (Ethical Application Ref: 64-556-19-2).

The following information was supplied regarding data availability:

The code is available at Zenodo: Kasikrit Damkliang. (2023). kasikrit/PCa-binary-segmentation: Version 1.0.1 (v1.0.1). Zenodo. https://doi.org/10.5281/zenodo.10141983.

The dataset and trained models are available at figshare: Damkliang, Kasikrit (2023). The 50 slides of dataset and trained models. figshare. Dataset. https://doi.org/10.6084/m9.figshare.24715626.v1.

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
