# Peer review of "Binary semantic segmentation for detection of prostate adenocarcinoma using an ensemble with attention and residual U-Net architectures"

_PeerJ Computer Science, doi:10.7717/peerj-cs.1767_

## Round 0.1 · original submission · Major Revisions

The reviewers have substantial concerns about this manuscript. The authors should provide point-to-point responses to address all the concerns and provide a revised manuscript with the revised parts being marked in different color.

Reviewer 1 ·

Basic reporting

The paper is well-written in clear and professional English. Appropriate background and context are provided in the introduction, with citations to relevant prior literature. The article structure generally conforms to standard scientific publication format with sections for Introduction, Related Works, Materials and Methods, Results and Discussions, and Conclusions. Figures are appropriately descriptive and informative. Tables effectively present key dataset characteristics and model evaluation metrics. The authors have helpfully provided the raw WSIs and ground truth masks as supplemental files. Overall, the manuscript is reasonably self-contained with presentation of an ensemble deep learning approach for prostate cancer tissue segmentation and evaluation on original histopathology data.

Some suggestions:

- Provide more details on the image acquisition and digitization - resolution, bit depth, scanner model etc.

- Expand the introduction to include more background on prostate cancer grading systems and clinical significance.

- The literature review could be expanded. For example, recent studies utilizing transformer-based models for medical image segmentation could be discussed.

- The authors should share the raw code and trained models in an open repository.

Experimental design

The research aims to develop an AI method for differentiating benign and malignant prostate tissue. The authors utilize an ensemble of attention-based U-Nets to segment prostate cancer histopathology images into benign and malignant classes. The research methodology is technically sound, leveraging many variant of U-Net architectures. Appropriate training, validation, testing, and comparison with other methods is performed. I have some suggestions for the authors regarding the experimental design.

Some suggestions:

- Provide more details on the biopsy sampling method and patient cohort. How were subjects selected? Were there any inclusion/exclusion criteria?

- More justification is needed for the dataset size of only 50 slides. This seems small to train a deep learning model without overfitting. Authors should discuss limitations of the dataset size.

- The technical methods used appear to follow standard practices. However, more details should be provided on the data annotation process and inter-observer variability between pathologists.

- The authors are encouraged to perform ablation studies to demonstrate the specific benefits of the attention gates and residual blocks added to the U-Net architecture.

Validity of the findings

The study does not make exaggerated claims of novelty or impact. The focus is on developing and evaluating a tailored solution for binary semantic segmentation of prostate histopathology images. The data appears sufficiently robust to support the conclusions. Appropriate data pre-processing and augmentation steps have been taken. The key conclusions relating to the model's ability to discriminate between benign and malignant prostate tissues are supported by the results on the testing dataset. The conclusions are limited to what can be supported by the data.

Some comments:

- How was the operating threshold selected for determining malignancy from the predicted probabilities?

- There is limited analysis of model performance on different Gleason grades or patterns. The malignant class grouped together GP3 and GP4, but it would be informative to see if the model performs differently on each grade.

- The model ensemble approach needs more justification - why specifically were these 3 models chosen and how complementary are they?

- Statistical significance testing is encouraged to demonstrate the ensemble model outperforms individual models.

Reviewer 2 ·

Basic reporting

The paper presents a commendable effort in automating Gleason Pattern (GP) segmentation in prostate adenocarcinoma, a critical aspect of prostate cancer diagnosis. The integration of an ensemble of U-Net-based architectures, along with the use of Attention Gate (AG) units and Residual Convolution Units, demonstrates a promising approach. However some points should be addressed before this manuscript is accepted.

Experimental design

Lines 94 and 163: It's crucial to maintain clarity in terminology. 'AUC' refers to 'Area Under the Curve,' while 'ROC' stands for 'Receiver Operating Characteristic.' Ensuring correct usage of these terms is important to avoid any confusion in the concept.

Line 231: Could you provide additional details on the ensemble method utilized in the study? More context regarding its implementation would enhance understanding.

Table 4: It might be more appropriate for Table 4 to be included within the supplementary materials for clarity and organization.

Validity of the findings

Discussion of Limitations: It would be valuable to include a section discussing any potential limitations of the study. Addressing these aspects would provide a more comprehensive assessment of the proposed method.

Additional comments

Acknowledgment Section: It seems that the acknowledgment section is incomplete or missing. Please ensure that all necessary acknowledgments are included and properly formatted.

Reviewer 3 ·

Basic reporting

Thanks for inviting me as one of reviewer. After checking the manuscript, ‘Semantic segmentation of gleason patterns in prostate adenocarcinoma exploiting an ensemble model of attention based U-Net’. I think this manuscript could be accepted after some problems are solved.

1.The selection of 50 WSIs and patients for this study should indeed be explained in more detail. The authors need to clarify why these specific patients were chosen and whether there were any criteria for patient selection. Additionally, any potential sample selection bias should be addressed in the manuscript.

2.The use of the GlaS-2015 dataset, which is unrelated to prostate cancer, as the external validation dataset raises a valid concern. It would be more appropriate to use a prostate cancer-specific dataset for validation to ensure the model's relevance to the target application. The authors should consider finding or using an appropriate prostate cancer dataset for validation.

3.The manuscript should provide more detailed information about the data preprocessing steps, especially regarding how outliers or low-quality images were handled. This information is essential for transparency and reproducibility.

4.The choice of image size (256 x 256 pixels) over 128 x 128 pixels should be justified. The authors should explain why this specific size was chosen and how it impacts the model's performance or computational requirements.

5.The manuscript should elaborate on how label consistency and reproducibility were ensured during the creation and adjustment of data labels. This is crucial for understanding the reliability of the ground truth annotations.

6.The authors should provide details on how the model's predictions are interpreted and whether the decision-making process of the model is understandable. Model interpretability is important, especially in medical applications.

7.While the model's performance in experiments is discussed, the manuscript should also address the feasibility and potential challenges of applying the model in a clinical setting. Considerations such as data acquisition in a clinical environment, real-time processing, and integration into clinical workflows should be discussed.

8.Code Availability: It is a good practice to make the code, especially for model construction, parameter tuning, and validation, available on platforms like GitHub to facilitate result replication and transparency. The authors should consider uploading their code for reproducibility purposes.

Experimental design

no comment

Validity of the findings

no comment

Additional comments

no comment

---

## Round 0.2 · accepted · Accept

Reviewers are satisfied with the revisions, and I concur to recommend accepting this manuscript.

Reviewer 1 ·

Basic reporting

The authors have addressed the questions in the comments

Experimental design

The authors have addressed the questions in the experimental design

Validity of the findings

The authors have addressed the questions in the validity of the findings

Reviewer 3 ·

Basic reporting

All concerns have been addressed satisfactorily by the authors, leading me to conclude that this manuscript is now suitable for publication.

Experimental design

All concerns have been addressed satisfactorily by the authors, leading me to conclude that this manuscript is now suitable for publication.

Validity of the findings

All concerns have been addressed satisfactorily by the authors, leading me to conclude that this manuscript is now suitable for publication.

Additional comments

All concerns have been addressed satisfactorily by the authors, leading me to conclude that this manuscript is now suitable for publication.